# Methicillin Resistance Increased the Risk of Treatment Failure in Native Joint Septic Arthritis Caused by *Staphylococcus aureus*

**DOI:** 10.3390/antibiotics12111628

**Published:** 2023-11-15

**Authors:** Jungok Kim, So Yeon Park, Kyung Mok Sohn, Bomi Kim, Eun-Jeong Joo

**Affiliations:** 1Division of Infectious Diseases, Department of Internal Medicine, Chungnam National University Sejong Hospital, Sejong 30099, Republic of Korea; kjo@cnuh.co.kr; 2Division of Infectious Diseases, Department of Internal Medicine, Chungnam National University School of Medicine, Deajeon 35015, Republic of Korea; medone@cnuh.co.kr; 3Division of Infectious Diseases, Department of Internal Medicine, Kangdong Sacred Heart Hospital, Seoul 05355, Republic of Korea; happyhappy@kdh.or.kr; 4Division of Infectious Diseases, Department of Medicine, Kangbuk Samsung Hospital, Seoul 03181, Republic of Korea

**Keywords:** arthritis, infectious, septic arthritis, native joint septic arthritis, bone and joint infections, methicillin-resistant *Staphylococcus aureus*

## Abstract

This study aimed to compare clinical characteristics and outcomes in patients with native joint septic arthritis (NJSA) due to methicillin-resistant *Staphylococcus aureus* (MRSA) in comparison to methicillin-sensitive *S. aureus* (MSSA) and identify treatment failure risk factors. We conducted a multi-center retrospective study on adult NJSA patients at three teaching hospitals in South Korea from 2005 to 2017. Among 101 patients diagnosed with *S. aureus* NJSA, 39 (38.6%) had MRSA strains. Compared to MSSA, patients with MRSA had a higher prevalence of nosocomial infections (17.9% vs. 1.6%; *p* = 0.005) and received inappropriate antibiotics within 48 h more frequently (74.4% vs. 0%; *p* < 0.001). In total, twenty patients (19.8%) experienced treatment failure, which encompassed five patients (5.0%) who passed away, nine (8.9%) requiring repeated surgical drainage after 30 days of antibiotic therapy, and seven (6.9%) with relapse. The MRSA group showed a higher rate of overall treatment failure (33.3% vs. 11.3%; *p* = 0.007) with a notably increased frequency of requiring repeated surgical interventions after 30 days of antibiotic therapy (17.9% vs. 3.2%, *p* = 0.026), in contrast to the MSSA group. Independent risk factors for treatment failure included Charlson comorbidity score, elevated CRP levels, and methicillin resistance. Methicillin resistance is an independent risk factor for treatment failure, emphasizing the need for vigilant monitoring and targeted interventions in MRSA-related NJSA cases.

## 1. Introduction

Septic arthritis in a native joint, characterized by pathogen invasion of joint spaces, is a severe and life-threatening condition that necessitates prompt diagnosis and effective management to prevent irreversible joint damage and severe complications [1]. *Staphylococcus aureus*, a prominent causative microorganism of native joint septic arthritis (NJSA), is responsible for a significant proportion of cases, ranging from 40% to 60% [2,3,4,5]. Global concern surrounds methicillin-resistant *S. aureus* (MRSA) as a leading public health threat [6,7,8]. MRSA introduces complexity to NJSA management, particularly due to its resistance, posing a formidable challenge for empirical antibiotic use. Therefore, a comprehensive understanding of pathogen characteristics and their relationship to host factors is imperative for NJSA management.

Patients infected with MRSA exhibit clinical distinctions compared to those with methicillin-susceptible *S. aureus* (MSSA). Additionally, MRSA infections are often associated with higher mortality rates and extended hospitalizations compared to MSSA infections [7]. Severe infections such as bacteremia or endocarditis have been associated with fatal outcomes [9,10,11,12,13,14]. Nonetheless, uncertainty remains regarding the impact of methicillin resistance on clinical results at specific infection sites. Previous research has established a correlation between MRSA and unfavorable outcomes in bone and joint infections [15,16]. However, many of these studies included diverse populations, encompassing not only NJSA but also cases with implanted prosthetic materials, osteomyelitis, or other bony structures. This diversity can affect outcomes through different management aspects. Due to the limited study of direct comparisons between MRSA and MSSA in NJSA cases, a knowledge gap exists concerning how MRSA clinically differs from MSSA and how MRSA affects prognosis in the context of NJSA [17,18,19].

A multicenter retrospective study was conducted to investigate clinical characteristics and outcomes of patients with NJSA caused by *S. aureus*, comparing MRSA to MSSA. We also assessed the impact of methicillin resistance on treatment failure in *S. aureus* NJSA.

## 2. Results

### 2.1. Demographic and Clinical Characteristics of S. aureus NJSA Compared between MRSA and MSSA 

A total of 101 patients with *S. aureus* NJSA were identified in three teaching hospitals during the study. Among them, 39 (38.6%) were MRSA cases. All patients identified microorganisms in joint fluid cultures, except for three MSSA cases who exclusively had a pathogen isolated from their blood. All patients who underwent surgical drainage via arthroscopy or arthrotomy were microbiologically confirmed by initial joint aspirations before surgical drainage, with the exception of one case that involved arthrotomy for a hand joint. The median age was 62 years, with males accounting for 56.4% of the cases. Comorbidities included diabetes mellitus (31.7%), osteoarthritis (16.8%), immunocompromised status (14.9%), and rheumatoid arthritis (5.9%). The knee was the most predominantly affected joint (57.4%), followed by the shoulder (18.8%) and hip (8.9%). Table 1 presents a comparison of demographics and clinical characteristics between the MSSA and MRSA groups. No significant differences were observed in the distribution of clinical presentations or affected joints between the two groups. Laboratory findings upon admission also showed similarities, including levels of CRP, ESR, and white blood cell count (WBC) in both serum and synovial fluid. However, hospital-acquired infection was more prevalent in the MRSA group (17.9% vs. 1.6%, *p* = 0.005).

### 2.2. Treatment and Outcomes in S. aureus NJSA Compared between MRSA and MSSA

Table 2 provides treatment and outcome details for *S. aureus* NJSA cases, comparing MRSA and MSSA. Initial drainage procedures were performed in 85.1% of patients, with 64.4% undergoing these procedures within 72 h of diagnosis. There were no differences in the modes and timing of drainage, duration of antibiotic therapy, or hospital length of stay (LOS) between the MRSA and MSSA groups. However, the MRSA group had a lower frequency of early drainage within 72 h (51.3% vs. 72.6%, *p* = 0.030) and appropriate antibiotics within 48 h (25.6% vs. 100%, *p* < 0.001) compared to the MSSA group. Additionally, they showed a tendency toward a higher occurrence of requiring secondary surgical debridement (25.6% vs. 17.7%, *p* = 0.451).

The median follow-up duration was 4 months (interquartile range [IQR], 2–24 months) during the study. In total, 20 (19.8%) patients experienced treatment failure, including five (5.0%) patients who passed away, nine (8.9%) who required repeated surgical drainage after 30 days while receiving antibiotic therapy, and seven (6.9%) with relapses. The MRSA group had a higher rate of overall treatment failure (33.3% vs. 11.3%, *p* = 0.007), involving an increased rate of requiring repeated surgical drainage (17.9% vs. 3.2%, *p* = 0.026), compared to the MSSA group. Although the MRSA group had a higher mortality rate (10.3% vs. 1.6%, *p* = 0.072), no significant differences were observed in the rates of death and relapse between the two groups. The Kaplan–Meier analysis of treatment success curves in NJSA, stratified by the presence of MRSA and MSSA, revealed a statistically significant difference (Figure 1).

### 2.3. Antibiotic Susceptibility

Table 3 describes antibiotic susceptibility profiles. All isolated *S. aureus* strains were sensitive to teicoplanin, vancomycin, tigecycline, and linezolid. Rifampin and trimethoprim/sulfamethoxazole exhibited high susceptibility rates, while fusidic acid had a low susceptibility rate. Compared to MSSA, MRSA displayed increased resistance to clindamycin, erythromycin, ciprofloxacin, and gentamicin. 

### 2.4. Risk Factors for Treatment Failure and the Need for Subsequent Surgical Drainage 

Risk factors for treatment failure in patients with NJSA caused by *S. aureus* are described in Table 4. According to univariate analysis, predisposing factors for treatment failure were the presence of the MRSA strain, higher CRP levels, Charlson comorbidity score, and patients on mechanical ventilators. In the multivariate analysis, elevated CRP levels and MRSA strain were identified as independent risk factors associated with treatment failure. In the subgroup analysis of treatment failure, predictors for the need for repeated surgical drainage after 30 days of antibiotic therapy included elevated CRP levels (odds ratio (OR), 1.12; 95% confidence interval (CI), 1.03–1.22; *p* = 0.010) and methicillin resistance (OR, 6.95; 95% CI, 1.24–38.94; *p* = 0.027), while elevated CRP levels (OR, 1.17; 95% CI, 1.01–1.35; *p* = 0.030) and the presence of shock (OR,19.19; 95CI, 1.58–232.02; *p* = 0.020) were associated with death. In the relapse subgroup, no variables were identified as prognostic factors in the multivariate analysis.

## 3. Discussion

In this study, we conducted a comparative analysis of clinical manifestations and treatment outcomes in patients with NJSA infected with MRSA and MSSA. Despite both groups exhibiting similar clinical presentations and disease severity, MRSA cases were significantly associated with hospital-acquired infections, delayed antibiotic administration, postponed initial drainage procedures, and a higher incidence of repeated surgical therapy after 30 days of antibiotic treatment, resulting in a higher rate of treatment failure. Independent risk factors for treatment failure in NJSA included elevated CRP levels and the presence of MRSA strains. This study holds significance in highlighting the adverse outcomes associated with MRSA in *S. aureus* NJSA cases, serving as a cautionary guide for cases with elevated CRP levels and methicillin resistance.

Our investigation revealed no discernible differences in clinical presentations, affected joints, or laboratory results between MRSA and MSSA cases of NJSA. This aligns with prior studies, which have found that distinguishing MRSA from MSSA based solely on clinical or initial laboratory parameters may be challenging [4,17,18,19]. However, the majority of hospital-acquired infections were attributed to MRSA, with only one exception. Furthermore, the delay in the administration of appropriate antibiotics (84.4%) was prevalent in NJSA by MRSA, as expected. Considering the importance of choosing the right antibiotics to maximize patient outcomes, these findings consistently support the critical relevance of empirical antibiotic selection for patients at risk of MRSA acquisition during hospitalization in epidemiology [2,8,19]. 

Previous studies have underscored the problematic implications of MRSA, often correlating it with elevated mortality rates [20,21,22]. However, the clinical ramifications of methicillin resistance in the context of bone and joint infections have yielded conflicting findings. In a comprehensive investigation of prosthetic joint infections (PJI), no significant prognostic disparity was discerned between MRSA and MSSA. Nonetheless, it was noted that MRSA PJI exhibited a higher rate of treatment failure during the treatment course, whereas treatment failure in MSSA cases occurred post-completion of antibiotic therapy [16]. Studies involving a mixed population of native joint and other types of bone and joint infections have reported similar outcomes for the MRSA and MSSA groups [13,15,18]. Conversely, another study on NJSA indicated higher fatality rates within the MRSA group [17,18,19]. In our study, MRSA NJSA demonstrated an elevated overall treatment failure rate when compared to MSSA NJSA, particularly necessitating repeated surgical interventions during the treatment course. Our research exclusively concentrated on NJSA cases, representing a distinctive subset of bone and joint infections. These findings offer a more focused perspective on the outcomes associated with MRSA NJSA, considering the unique therapeutic approaches required for NJSA in contrast to other types of bone and joint infections. Given the increased frequency of additional surgical interventions in the MRSA group, it is imperative to maintain vigilant monitoring for clinical signs and data suggestive of treatment failure during the treatment of MRSA NJSA. Furthermore, this observation underscores the pivotal role of a multidisciplinary approach in enhancing the management of septic arthritis, with close collaboration among specialists from various fields. 

In clinical practice, patients occasionally experience repeated operative interventions during antibiotic treatment, either due to deteriorating conditions or a lack of progress, even if they initially showed clinical improvement with reduced inflammatory markers [23,24,25]. Therefore, we classified this occurrence as a subcategory of treatment failure. *S. aureus* strains may serve as potential prognostic indicators for the inadequacy of a single debridement procedure [13,24]. Given the delayed administration of appropriate antibiotics in MRSA cases, we hypothesized a possible exacerbation of joint inflammation in MRSA NJSA. Our analysis revealed the adverse effect of the MRSA strain itself on treatment outcomes, especially in terms of repeated surgical intervention, along with elevated CRP level. Previous research suggests that predictors differ between treatment failure and mortality in septic arthritis [26]. Local factors and systemic complications tend to predict treatment failure, while host-related conditions such as age and comorbidities are generally more predictive in terms of mortality [26,27,28,29]. Consequently, predictors may show variability depending on the prognostic evaluation measure. However, our small sample size precluded us from drawing definitive conclusions regarding predictors for each subgroup of treatment failure.

Interestingly, we did observe a lower portion of patients undergoing drainage within 72 h in the MRSA group, even though the total time to drainage did not significantly differ between the two groups. Moreover, our univariate analysis did not identify drainage within 72 h as a significant favorable factor for treatment failure or repeated surgical treatment. Previous studies have reported that immediate surgical drainage does not appear to prevent long-term joint-related sequelae if patients are already receiving empiric antibiotic treatment, and it does not increase the risk of mortality [30,31]. While early drainage is recommended for infection control in septic arthritis, further exploration is needed to examine the impact of the optimal timing of surgery on treatment outcomes [25]. 

In this study, MRSA comprised 38.6% of *S. aureus* isolates, and this distribution was consistent with antibiotic susceptibility profiles in South Korea, which have been reported to vary from 33 to 77% in different regions [3,32,33]. Approximately one-third of MRSA cases exhibited resistance to both ciprofloxacin and clindamycin. Conversely, linezolid and trimethoprim/sulfamethoxazole displayed high susceptibility. Current NJSA guidelines recommend an initial phase of parenteral therapy with the option to transition to oral therapy if the patient’s clinical condition improves, although the exact timing for this transition has been a subject of ongoing debate [34,35,36]. In MRSA cases, switching to oral antimicrobial agents is often restricted due to their resistance patterns and the necessity of combination therapy with specific susceptible agents [37,38]. Despite no difference in the duration of intravenous or oral agent use between the MRSA and MSSA cases in this study, the resistant pattern in MRSA NJSA cases contributes valuable epidemiological data and supplementary information for selecting oral agents or empirical antibiotics in cases where MRSA is suspected in treatment strategies. 

Several limitations must be considered when interpreting the results. First, the relatively small sample size, especially when stratified into MRSA and MSSA groups, may limit the generalizability of our findings. Second, the retrospective design of this study inherently carries the risk of encountering incomplete data, potential selection bias, and limited control over data quality. Despite our efforts to reconcile conflicting data, there may be unaccounted-for variables and missing details that could impact the study’s outcomes. Additionally, data collected across multiple hospitals might produce variations in data recording and reporting practices, although we utilized standardized data collection forms and cross-verified the data. Third, management practices can vary based on clinician experience, institutional protocols, surgical techniques, and patient-specific factors among different hospitals, which could serve as confounding factors. Lastly, we did not examine the specifics of antibiotic selection due to its heterogeneity, making it difficult to compare and evaluate outcomes in this regard. Nonetheless, all hospitals implement an online antibiotic prescription system under the oversight of antibiotic stewardship programs, so we assumed that most patients received the proper antibiotics at the correct dosage.

In conclusion, MRSA septic arthritis in native joints exhibits clinical similarities to MSSA septic arthritis but leads to unfavorable outcomes. Methicillin resistance was also identified as an independent risk factor for treatment failure. Clinicians should be alert in recognizing the risk factors, and early identification of MRSA pathogens, especially among patients with hospital-acquired infections, is crucial for making better choices regarding appropriate antibiotic administration in NJSA. The observed high rates of treatment failure among MRSA cases during the treatment period emphasize the paramount role of vigilant monitoring and prompt intervention for treatment success, as well as the necessity for a multidisciplinary approach to managing NJSA. Further research is warranted to refine prognostic factors that impact patient care.

## 4. Materials and Methods

### 4.1. Study Population, Design, and Data Collection 

A retrospective study was conducted from January 2005 to December 2017 to investigate the disparities between MRSA and MSSA in NJSA across three teaching hospitals. Patients were screened via synovial fluid culture results in the microbiology database. Eligible patients met the NJSA diagnostic criteria outlined by Newman [39]: aged ≥18 years, with *S. aureus* isolated from synovial fluid, and/or a discharge diagnosis of infectious arthropathy [11]. Exclusion criteria included patients with culture-negative septic arthritis, isolation of microorganisms other than *S. aureus*, presence of joint prosthetics, and suspected or confirmed osteomyelitis (e.g., in diabetes mellitus foot ulcers), and polymicrobial infections.

Patient medical records were meticulously reviewed using structured forms at each hospital. Data extracted for each patient included: age, gender, comorbidities, clinical presentations, hospitalization history, laboratory findings upon admission, susceptibility test results, administered antimicrobial agents, drainage procedures, surgical interventions, hospitalization duration, and patient outcomes. Severity of illness was assessed using the Charlson comorbidity score [40]. Drainage methods comprised repeated arthrocentesis and surgical debridement via arthroscopy or arthrotomy. The number of surgical debridement procedures was noted for cases where patients underwent surgery due to persistent infection during concurrent antibiotic therapy but not for cases related to joint function improvement through arthroplasty, arthrodesis, or amputation. Medical records underwent thorough scrutiny, with discrepancies cross-checked by an additional author.

### 4.2. Outcome and Follow-Up

Outcome was categorized as treatment success or treatment failure. Treatment success involved the resolution of infection-related signs and symptoms by therapy completion, irrespective of residual joint dysfunction sequelae. Treatment failure was categorized as follows: (i) “death”, involving cases directly or indirectly linked to NJSA within 30 days; (ii) “repeated surgical drainage after 30 days of antibiotic therapy”, representing cases necessitating further surgical drainage after 30 days due to persistent infection during receiving antibiotics, and (iii) “relapse”, indicating the reappearance of symptoms in previously affected joints due to reinfection or persistency with isolated microorganisms after completing antibiotic therapy. Subcategory of (ii) was corroborated by clinical signs, symptoms, and elevated acute-phase reactants such as C-reactive protein (CRP) or erythrocyte sedimentation rate (ESR) after initial improvement, or isolation of the same microorganisms in new samples during treatment.

### 4.3. Statistical Analysis

Continuous variables were analyzed using the Student *t*-test and Mann–Whitney test, while categorical variables were compared using the χ2 and Fisher’s exact tests. The cumulative rate of treatment failure-free survival was estimated using the Kaplan–Meier method, and treatment failure equality among groups was assessed with the log-rank test. To control for potential confounding variables, stepwise logistic regression analysis was performed. Variables with *p* < 0.1 in univariate analysis were included in the multivariate analysis to identify independent risk factors of treatment failure. Data were analyzed using SPSS version 26.0 (IBM Corp., Armonk, NY, USA), with statistical significance considered at *p* < 0.05 (two-tailed).

## Figures and Tables

**Figure 1 antibiotics-12-01628-f001:**
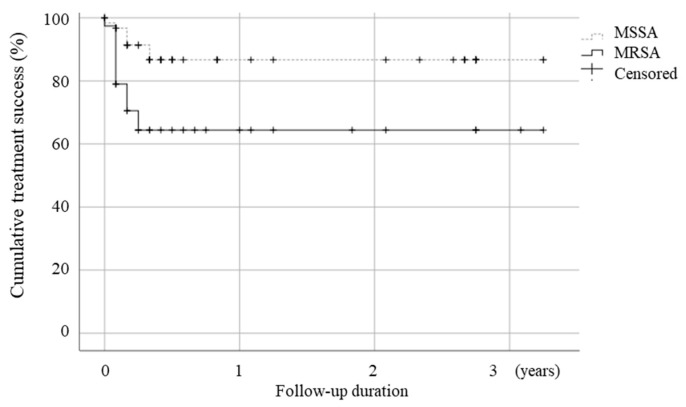
Kaplan–Meier analysis of treatment failure-free in native joint septic arthritis (*p* = 0.05).

**Table 1 antibiotics-12-01628-t001:** Clinical characteristics of patients with native joint septic arthritis caused by *S. aureus*.

	MRSA (*n* = 39 [38.6])	MSSA (*n* = 62 [61.4])	Total (*n* = 101)	*p*-Value
Age ≥ 65 years	18 (46.2)	26 (41.9)	44 (43.6)	0.677
Male	21 (53.8)	36 (58.1)	57 (56.4)	0.677
Comorbidities				
Immunocompromised status	5 (12.8)	10 (16.1)	15 (14.9)	0.788
Solid tumor	2 (5.1)	2 (3.2)	4 (4.0)	1.000 ^a^
Hematologic malignancy	1 (2.6)	1 (1.6)	2 (2.0)	1.000 ^a^
Immunosuppressant agents	4 (10.3)	9 (14.5)	13 (12.9)	0.565
End-stage renal disease	3 (7.7)	2 (3.2)	5 (5.0)	0.372 ^a^
Liver cirrhosis	1 (2.6)	4 (6.5)	5 (5.0)	0.646 ^a^
Diabetes mellitus	9 (23.1)	23 (37.1)	32 (31.7)	0.188
Rheumatoid arthritis	2 (5.1)	4 (6.5)	6 (5.9)	1.000 ^a^
Osteoarthritis	5 (12.8)	12 (19.4)	17 (16.8)	0.430
Charlson comorbidity score ^b^	1.08 ± 1.56	1.02 ± 1.52	1.04 ± 1.52	0.847
Risk factors				
Previous intra-articular injection	12 (30.8)	19 (30.6)	31 (30.7)	1.000
Previous acupuncture	1 (2.6)	6 (9.7)	7 (6.9)	0.244 ^a^
Previous arthroscopic procedure	6 (15.4)	6 (9.7)	12 (11.9)	0.529 ^a^
Recent blunt trauma	8 (20.5)	15 (24.2)	23 (22.8)	0.809
Hospital-acquired infection	7 (17.9)	1 (1.6)	8 (7.9)	0.005 ^a^
Involved joints				
Knee	22 (56.4)	36 (58.1)	58 (57.4)	1.000
Hip	3 (7.7)	7 (11.3)	10 (8.9)	0.737 ^a^
Shoulder	9 (23.9)	10 (16.1)	19 (18.8)	0.438
Elbow	4 (10.3)	3 (4.8)	7 (6.9)	0.425 ^a^
Ankle	1 (2.6)	3 (4.8)	4 (4.0)	1.000 ^a^
Wrist	1 (2.6)	2 (3.2)	3 (3.0)	1.000 ^a^
Other small joints	0 (0)	3 (4.8)	3 (3.0)	0.282 ^a^
Polyarthropathy	1 (2.6)	2 (3.2)	3 (3.0)	1.000 ^a^
Clinical presentation				
Body temperature ≥ 38 ℃	14 (35.9)	29 (46.8)	43 (42.6)	0.308
Shock	3 (7.7)	5 (8.1)	8 (7.9)	1.000 ^a^
Mechanical ventilator	3 (7.7)	3 (4.8)	6 (5.9)	0.674 ^a^
Laboratory test at admission				
WBC, ×10^9^/mm^3^	11.5 (7.1–14.1)	11.9 (10.-14.3)	11.7 (9.5–14.1)	0.619
CRP, mg/dL	13.3 (0.1–42.0)	13.5 (0.1–34.4)	12.8 (7.7–21.4)	0.399
ESR, mm/h	81 (8–120)	75 (7–124)	77 (50–109)	0.507
Acute kidney injury	5 (12.8)	8 (12.9)	13 (12.9)	1.000
Synovial fluid WBC, ×10^3^/mm^3^	55.1 (35.6–111.75)	113.1 (38.9–175.7)	91.5 (37.8–158.5)	0.182
Positive blood culture ^c^	13 (54.2)	26 (57.8)	39 (56.5)	0.803

The data are expressed as numbers (%), unless otherwise indicated. Continuous variables are expressed as median and interquartile range (IQR). MRSA, methicillin-resistant *S. aureus*; MSSA, methicillin-sensitive *S. aureus*; WBC, white blood cell count; CRP, C-reactive protein; ESR, erythrocyte segmentation rate. ^a^ Fisher’s exact test ^b^ The Charlson comorbidity score is presented as mean and standard deviation. ^c^ The proportion was calculated based on 69 cases of blood culture.

**Table 2 antibiotics-12-01628-t002:** Treatment and outcomes of patients with native joint septic arthritis caused by *S. aureus*.

	MRSA (*n* = 39 [38.6])	MSSA (*n* = 62 [61.4])	Total (*n* = 101)	*p*-Value
Initial drainage modes	33 (84.6)	53 (85.5)	86 (85.1)	1.000
Time to drainage, days	2.0 (0–4)	1.0 (0–2)	1.0 (0–3)	0.093
Drainage ≤ 72 h	20 (51.3)	45 (72.6)	65 (64.4)	0.030
Repeated arthrocentesis	4 (10.3)	10 (16.1)	14 (13.9)	0.557
Arthroscopy	21 (53.8)	38 (61.3)	59 (58.4)	0.536
Arthrotomy	8 (20.5)	5 (8.1)	13 (12.9)	0.124
Surgical drainage ≥ 2 times ^b^	10 (25.6)	11(17.7)	21 (20.8)	0.451
Appropriate antibiotics ≤ 48 h	10 (25.6)	62 (100)	72 (71.3)	<0.001
Duration of antibiotic therapy, days				
Total antibiotics	37 (25–52)	47 (33–69)	44 (30–64)	0.238
≤4 weeks	13 (33.3)	12 (19.4)	25 (24.8)	0.113
4–6 weeks	9 (23.1)	13 (21.0)	21 (21.8)	0.803
>6 weeks	17 (43.6)	37 (59.7)	54 (53.5)	0.115
Intravenous antibiotics	29 (22–39)	28 (21–41)	28 (22–40)	0.692
Oral antibiotics ^c^	0 (0–21)	13 (0–31)	12 (0–28)	0.166
Hospital length of stay, days	41 (31–74)	34 (24–52)	36 (25–59)	0.266
Treatment failure	13 (33.3)	7 (11.3)	20 (19.8)	0.007
Death	4 (10.3)	1 (1.6)	5 (5.0)	0.072 ^a^
Repeated surgical drainage after 30 days of antibiotic therapy ^d^	7 (17.9)	2 (3.2)	9 (8.9)	0.026 ^a^
Relapse after completed therapy ^e^	2 (5.1)	5 (8.1)	7 (6.9)	0.704 ^a^

The data are expressed as numbers (%), unless otherwise indicated. Continuous variables are expressed as median and interquartile range (IQR). MRSA, methicillin-resistant *S. aureus*; MSSA, methicillin-sensitive *S. aureus*
^a^ Fisher’s exact test ^b^ Cases underwent a second surgical drainage procedure with a median time interval of 22 days after the initial procedure (ranging from 5 to 79 days). Four patients underwent three surgical procedures, including three MRSA cases and one MSSA case. ^c^ Twenty-two patients with MRSA remained on intravenous therapy and did not switch to oral agents at the end of therapy. ^d^ The median time was 35 days, ranging from 33 to 79 days. ^e^ The interval time to relapse ranged from 9 to 38 days, with the exception of one MSSA case who experienced repeated surgical drainage at 79 days of antibiotic therapy and then relapsed 2 years after completing therapy.

**Table 3 antibiotics-12-01628-t003:** Antibiotic susceptibility of *S. aureus* in native joint septic arthritis.

Antimicrobial Agents	Number of Susceptible Isolates (%)			*p*-Value
	MRSA (*n* = 39)	MSSA (*n* = 62)	Total (*n* = 101)	
Clindamycin ^b^	19 (48.7)	51 (82.3)	70 (69.3)	<0.001
Erythromycin	17 (43.6)	50 (80.6)	67 (66.3)	<0.001
Trimethoprim/sulfamethoxazole	39 (100)	61 (98.4)	100 (99)	1.000 ^a^
Rifampin	39 (100)	61 (98.4)	100 (99)	1.000 ^a^
Ciprofloxacin ^b^	27 (69.2)	61 (98.4)	88 (87.1)	<0.001 ^a^
Fusidic acid	36 (76.9)	42 (67.7)	72 (71.3)	0.564
Gentamicin	27 (69.2)	59 (95.2)	86 (85.1)	<0.001
Penicillin	0 (0)	10 (16.1)	10 (9.9)	0.006 ^a^

The data are expressed as numbers (%), unless otherwise indicated. Continuous variables are expressed as median and interquartile range (IQR). All isolated *S. aureus* strains were sensitive to vancomycin, teicoplanin, tigecycline, and linezolid. MRSA, methicillin-resistant *S. aureus*; MSSA, methicillin-sensitive *S. aureus*
^a^ Fisher’s exact test ^b^ Twelve MRSA isolates displayed resistance to both ciprofloxacin and clindamycin.

**Table 4 antibiotics-12-01628-t004:** Risk factors for treatment failure in patients with native joint septic arthritis caused by *S. aureus*.

Variable	Treatment Success (*n* = 81 [80.1])	Treatment Failure (*n* = 20 [19.8])	Univariate Analysis		Multivariate Analysis	
			OR (95% CI)	*p*-Value	OR (95% CI)	*p*-Value
MRSA	26 (32.1)	13 (65.0)	3.93 (1.40–11.01)	0.007	3.38 (1.09–10.46)	0.035
Initial CRP (mg/dL)	13.1 ± 8.3	21.5 ± 10.0	1.10 (1.04–1.17)	0.002	1.09 (1.02–1.16)	0.007
Charlson comorbidity score	0.85 ± 1.53	1.8 ± 1.28	1.41 (1.04–1.90)	0.008		
Mechanical ventilator	3 (3.7)	3 (15)	4.59 (0.85–24.72)	0.076		

Variables with *p* < 0.1 were included in the multivariate analysis. Data are expressed as no. (%) of patients, unless otherwise indicated. Continuous variables expressed as mean and standard deviation. OR, Odds ratio; CI, confidence interval; MRSA, methicillin-resistant *S. aureus*; CRP, C-reactive protein.

## Data Availability

Data are contained within the article.

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
