# Peer review of "Methicillin Resistance Increased the Risk of Treatment Failure in Native Joint Septic Arthritis Caused by Staphylococcus aureus"

_antibiotics, 2023, doi:10.3390/antibiotics12111628_

Round 1

Reviewer 1 Report

Comments and Suggestions for Authors

The authors undertook the Methicillin Resistance Increased the Risk of Treatment Failure in Native Joint Septic Arthritis caused by Staphylococcus aureus. The manuscript is important for clinician for better treatments. 

What about the other regions??? 

What about the solution of this issue, authors need to address. 

Reviewer 2 Report

Comments and Suggestions for Authors

The Authors aimed to compare clinical characteristics and outcomes in patients with native joint septic arthritis (NJSA) due to methicillin-resistant Staphylococcus aureus (MRSA) in comparison to methicillin-sensitive S. aureus (MSSA), and identify treatment failure risk factors.

The topic is interesting and the study well designed.

Did any of the patients receive joint aspiration before surgery?Did this modify treatment?

Was procalcitonin tested and used as an urgency marker?

Timing of surgery differed between the two groups. This might have affected results. Please discuss

Reviewer 3 Report

Comments and Suggestions for Authors

The main questions addressed by the research was investigated clinical characteristics and outcomes of patients with native joint septic arthritis (NJSA) due to methicillin resistant Staphylococcus aureus (MRSA) in comparison to methicillin sensitive S. aureus (MSSA), and identify treatment failure risk factors. The authors conducted a multicenter retrospective study on adult NJSA patients at three teaching hospitals in South Korea from 2005 to 2017. I consider the topic is oryginal and relevant in this field because the researches concentrated on NJSA cases, representing a distinctive subset of bone and joint infections. 
